# Direct observation of accelerating hydrogen spillover via surface-lattice-confinement effect

Yijing Liu[1,2,5], Rankun Zhang[1,3,5], Le Lin[1,5], Yichao Wang[2,4], Changping Liu[1,2], Rentao Mu [1]✉ & Qiang Fu [1]✉

Uncovering how hydrogen transfers and what factors control hydrogen conductivity on solid surface is essential for enhancing catalytic performance of H-involving reactions, which is however hampered due to the structural complexity of powder catalysts, in particular, for oxide catalysts. Here, we construct stripe-like MnO(001) and grid-like $Mn_3O_4$(001) monolayers on Pt(111) substrate and investigate hydrogen spillover atop. Atomic-scale visualization demonstrates that hydrogen species from Pt diffuse unidirectionally along the stripes on MnO(001), whereas it exhibits an isotropic pathway on $Mn_3O_4$(001). Dynamic surface imaging in $H_2$ atmosphere reveals that hydrogen diffuses 4 times more rapidly on MnO than the case on $Mn_3O_4$, which is promoted by one-dimension surface-lattice-confinement effect. Theoretical calculations indicate that a uniform and medium O-O distance favors hydrogen diffusion while low-coordinate surface O atom inhibits it. Our work illustrates the surface-lattice-confinement effect of oxide catalysts on hydrogen spillover and provides a promising route to improve the hydrogen spillover efficiency.

The "hydrogen spillover", first evidenced in experiments by Khoobiar in 1964[1], depicts the dynamic migration of surface adsorbed hydrogen species from hydrogen-rich sites to hydrogen-poor sites. Considering its great potential in H-involving reaction processes, including methanol synthesis[2], Fischer-Tropsch synthesis[3], hydrogenations[4,5], hydrogen storage[6], etc., hydrogen spillover comes to a hot research topic among scientists, who intend not only to interpret it but also to exploit it for improving reaction performance and functionalizing materials[7,8]. In many cases, the migration of spilt hydrogen atoms is determined to be the rate-determining step in hydrogenation reactions. For example, the higher diffusion rates of the hydrogen species on chromium oxide than that on zinc or aluminum oxides corresponds to the following hydrogenation rates order: $Cr_2O_3 \gg ZnO \approx Al_2O_3$[9]. For spatially separated Pt and Co nanoparticles on $SiO_2$, the hydrogen

atoms can diffuse across the $SiO_2$ support to reduce surface oxygen-containing species on Co nanoparticles, and thus promoting $CO_2$ methanation reaction[10]. Tan et al. also found that an enhanced hydrogen spillover from Pt to Fe over the $SiO_2$ support with the assistance of gaseous oxygenate molecules containing carbonyl functional group significantly accelerates the rate of hydrodeoxygenation of pyrolysis bio-oil vapor[11].

Although hydrogen spillover has been implicated in a variety of scientific and technological fields, it has proven to be challenging to demonstrate the dynamics and kinetics of spilt hydrogen species. The blossoming of in situ characterization technologies has pushed forward the monitoring of hydrogen spillover[8]. By using X-ray photoelectron spectroscopy (XPS), the changes of surface oxidation states during hydrogen spillover on oxides have been investigated[12].

[1]State Key Laboratory of Catalysis, Dalian Institute of Chemical Physics, Chinese Academy of Sciences, 116023 Dalian, China. [2]University of Chinese Academy of Sciences, 100039 Beijing, China. [3]Zhang Dayu School of Chemistry, Dalian University of Technology, 116024 Dalian, China. [4]CAS Key Laboratory of Science and Technology on Applied Catalysis, Dalian Institute of Chemical Physics, Chinese Academy of Sciences, 116023 Dalian, China. [5]These authors contributed equally: Yijing Liu, Rankun Zhang, Le Lin. ✉e-mail: murt@dicp.ac.cn; qfu@dicp.ac.cn

However, tracking hydrogen spillover at nanoscale is highly demanding in order to reveal the active sites. Employing nanometer-scale resolved X-ray absorption spectroscopy in a X-ray photoemission electron microscope, Karim et al.[13] concluded that hydrogen spillover on $Al_2O_3$ is slower and limited to shorter distance than that on reducible $TiO_2$. Tip-enhanced Raman spectroscopy combined with scanning tunneling microscopy (STM) has been employed to study hydrogen spillover at nanoscale[14,15]. Taking advantage of low-temperature STM, the diffusion of hydrogen atoms from surface Pd sites to surrounding Cu(111) and Ag(111) surface domains on bimetallic catalysts has been directly observed[16,17]. However, microscopic understanding of the intrinsic properties of oxide surfaces that determine the hydrogen spillover process is still missing, thus calling for in situ/operando surface characterizations in $H_2$ atmosphere and at atomic scale.

Manganese oxides ($MnO_x$) have been widely utilized in H-involving reaction processes, such as hydrogenation[18] and hydrogen storage[19], during which $MnO_x$ with different structures usually exhibit divergent performance.

Here, we constructed two well-defined $MnO_x$ surfaces, stripe-like MnO(001) and grid-like $Mn_3O_4$(001) monolayers on Pt(111) substrate, whose surface structures differ immensely from each other. High pressure STM (HP-STM) and XPS experiments reveal that the spillover direction is effectively regulated by the surface structure and the rate is largely promoted by the one-dimension (1D) surface-lattice-confinement effect. Density functional theory (DFT) calculations demonstrate that the differences of hydrogen diffusion barrier are essentially related to the local surface geometries and coordination numbers of surface O sites in the Mn oxide monolayers.

## Results

### Construction of monolayer MnO(001) and $Mn_3O_4$(001)

$MnO_x$ nanoislands were prepared by reactive evaporation of Mn in $O_2$ atmosphere on Pt(111) substrate. Depositing Mn atoms in

$1 \times 10^{-7}$ mbar $O_2$ at 423 K followed by annealing in vacuum to 700 K produces 0.6 monolayer (ML) $MnO_x$ islands with ~1.8 Å apparent height. The island surface exhibits a characteristic uni-axial stripe structure with an averaged distance of ~5.7 Å ([$2\bar{1}\bar{1}$] direction) and a corrugation of 2.8 Å in each stripe ([$01\bar{1}$] direction). As indicated by black dashed circles in Fig. 1a, a small number of dark spots, which might be oxygen vacancies, are distributed randomly on the island surface. Closer inspection of the atomic periodicity indicates that the stripes are in groups of 3 or 2 (a 3-2-3 sequence is shown in Fig. 1b), between which a shift close to half a unit cell exists along [$01\bar{1}$] direction. The atomic STM image inserted in Fig. 1b indicates that each stripe consists of two Mn or O rows, which is consistent with the previously reported structure of MnO(001)/Pt(111) surface[20]. Accordingly, the atomic model structure is proposed in Fig. 1c. While depositing Mn atoms in $5 \times 10^{-7}$ mbar $O_2$ at 373 K followed by annealing in vacuum to 600 K, a grid-like surface structure (Fig. 1d) with ~2.0 Å apparent height and 0.7 ML coverage can be obtained. The atomic structure (Fig. 1e) is almost the same as that of $Mn_3O_4$/Au(111) surface prepared in "oxygen rich" regime[21], where the bright features are arranged in a zigzag-like pattern with a spacing of ~3.0 Å.

The bottom curves in Fig. 2a, b present XPS O 1s spectra of as-prepared stripe-like and grid-like $MnO_x$/Pt(111) surfaces, respectively. O 1s peak of the as-prepared stripe-like $MnO_x$ (Fig. 2a) is located at 529.9 eV, which is consistent with the peak position of lattice O ($O_L$) of MnO[22]. Then, the XPS O 1s/Mn 2p signal ratio from the MnO(001)/Pt(111) surface was normalized to 1.00. O 1s peak of the as-prepared grid-like $MnO_x$ (Fig. 2b) is located at 529.7 eV, 0.2 eV lower than that of MnO, which can be assigned to $O_L$ of $Mn_3O_4$[22]. The XPS O 1s/Mn 2p signal ratio of the grid-like $MnO_x$ is determined to be 1.31. Accordingly, the grid-like surface can be assigned to distorted $Mn_3O_4$(001) on Pt(111). Its model is proposed in Fig. 1f.

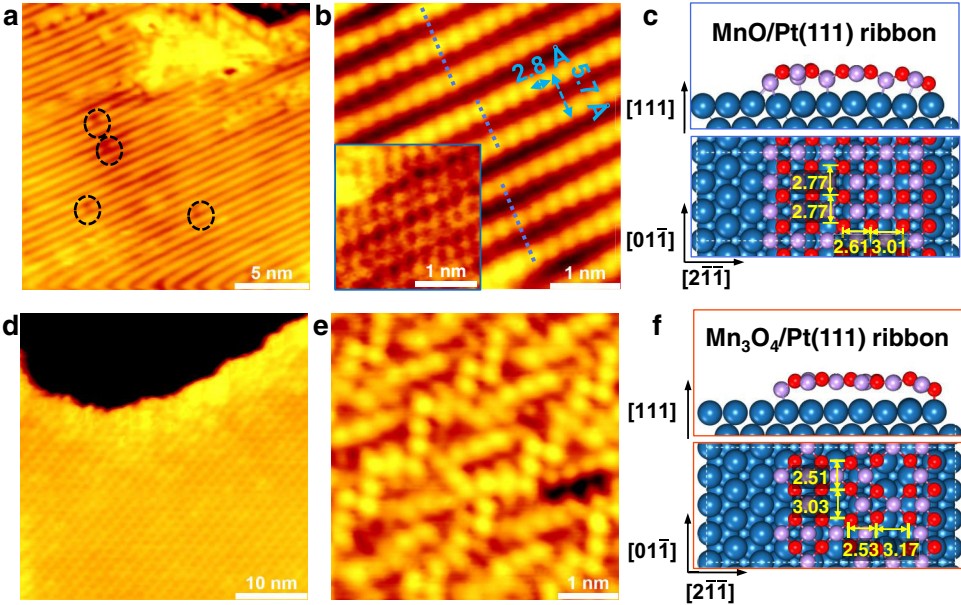

**Fig. 1 | STM images and proposed models of manganese oxides on Pt(111).**
**a** STM image of as-prepared stripe-like MnO(001)/Pt(111) surface. **b** Atomic corrugation of stripe-like MnO(001) and the inset in (**b**) shows the atomic-resolution STM image of the surface. **c** A (6 × 3) MnO ribbon supported on the (5√3 × 3) Pt(111) substrate, which is equivalent to a moiety from the reported (19 × 1) reconstruction. The orthogonal MnO monolayer can be truncated along the rock-salt MnO(001) surface. **d** STM image of as-prepared grid-like $Mn_3O_4$(001)/Pt(111) surface. **e** Atomic-resolution STM image of the grid-like $Mn_3O_4$(001). **f** An $Mn_3O_4$

ribbon supported on the (5√3 × 4) Pt(111) substrate. The $Mn_3O_4$ monolayer is derived from a reconstruction of the spinel $Mn_3O_4$(001) truncation. Scanning parameters: (**a**, **b**) $I_t$ = 0.090 nA, $V_s$ = 0.969 V; inset in (**b**) $I_t$ = 0.280 nA, $V_s$ = 0.009 V; (**d**) $I_t$ = 0.100 nA, $V_s$ = 1.400 V; (**e**) $I_t$ = 0.240 nA, $V_s$ = 0.008 V. O: red; Mn: light violet; Pt: dark blue. Note that the distances denoted by yellow numbers are obtained by correcting the calculated values (2.82 Å) to align to the experimental Pt lattice constant (2.77 Å).

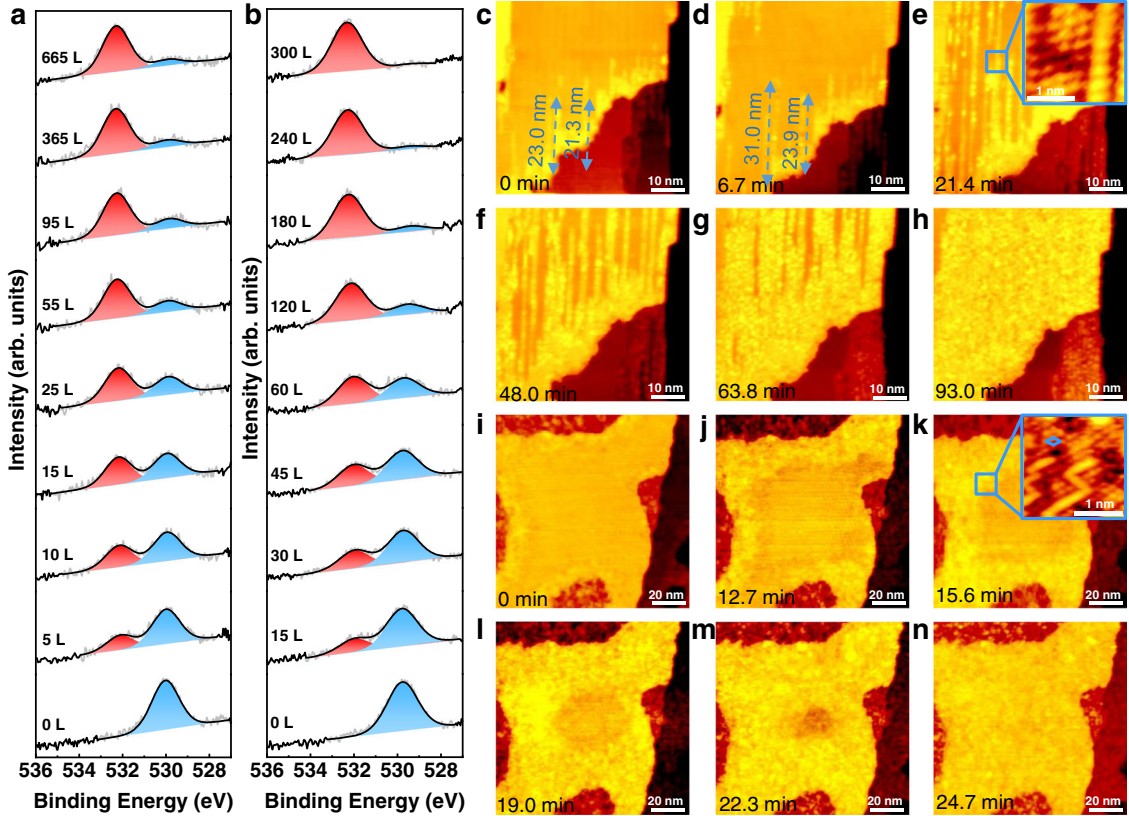

**Fig. 2 | The spillover processes on stripe-like MnO/Pt(111) and grid-like Mn₃O₄/Pt(111) surfaces. a, b** XPS O 1s spectra of stripe-like MnO and grid-like Mn₃O₄ surfaces exposed to different amount of D₂ at room temperature. A time-lapse sequence of STM images of (**c–h**) stripe-like MnO and (**i–n**) grid-like Mn₃O₄ in H₂ atmosphere at room temperature. Insets in (**e**) and (**k**) show atomic-resolution STM image of partially hydroxylated stripe-like MnO surface and partially hydroxylated grid-like Mn₃O₄ surface, respectively. The partial pressure of H₂ atmosphere from (**d**) to (**h**) is ~2 × 10⁻⁸ mbar, and from (**j**) to (**n**) is 2.5 × 10⁻⁷ mbar. The exposure time is stamped in the lower left corner. Scanning parameters: (**c, d**) $I_t = 0.090$ nA, $V_s = 0.831$ V; (**e–h**) $I_t = 0.090$ nA, $V_s = 0.937$ V; inset in (**e**) $I_t = 0.360$ nA, $V_s = 0.008$ V; (**i**) $I_t = 0.100$ nA, $V_s = 1.030$ V; (**j–n**) $I_t = 0.110$ nA, $V_s = 1.030$ V; inset in (**k**) $I_t = 0.100$ nA, $V_s = 0.019$ V.

## Hydrogen spillover on stripe-like MnO(001) and grid-like Mn₃O₄(001) in H₂

XPS O 1s spectra of stripe-like MnO surface exposed to increasing amount of D₂ are shown in Fig. 2a. After the sample is exposed to 5 L D₂, a new peak located at 532.1 eV emerges, which can be assigned to OD[23]. The peak area of O_L decreases by 86% while that of OD increases when the D₂ exposure amount increases from 0 to 365 L, indicating the transformation of O_L to OD. When continuing to increase the D₂ dosing to 665 L, the peak areas of OD and O_L remain almost unchanged, with their peak positions located at 532.3 and 529.6 eV, respectively. The maximum hydroxylation degree is calculated to be 88%, implying that MnOD_x is formed after the exposure of D₂ to MnO[24].

HP-STM was employed to investigate the hydroxylation process in situ. For the as-prepared stripe-like MnO surface, a few bright lines with ~1.1 Å apparent height exist on the surface as kept in UHV (Fig. 2c), which might be induced by H₂ in the background. The hydroxylation process was further investigated under 2.0 × 10⁻⁸ mbar H₂. Most of these bright lines start from the edge of the MnO island and extend to the middle following the shorter cell vector [01$\bar{1}$] direction. The bright lines continue to grow along 1D pathway with the increasing H₂ dosage (Fig. 2c–h, for the full series see Supplementary Movie 1). As indicated by the blue dashed lines from Fig. 2c, d, the length changes of each bright line vary, indicating that the spillover rates among all lines are different, which might result from the drag of oxygen vacancies on the hydrogen diffusion. Inset in Fig. 2e implies that the hydroxylation of stripe-like MnO surface is accompanied by surface reconstruction from the tetragonal to hexagonal symmetry.

In order to investigate the origin of hydroxylation, a 1.2 ML stripe-like MnO overlayer was grown for comparison (Supplementary Fig. 1a). Then, the sample was exposed to 1 × 10⁻⁶ mbar H₂ and no bright lines appeared on the surface (Supplementary Fig. 1b), indicating that hydroxylation cannot take place without bare Pt surface. Therefore, it can be inferred that H₂ dissociates on bare Pt region of the sub-monolayer MnO/Pt(111) surfaces and then dissociative hydrogen atoms spillover from Pt to MnO islands[25].

XPS O 1s spectra of grid-like Mn₃O₄ exposed to D₂ are shown in Fig. 2b. Upon the exposure of 15 L D₂, a new peak appears at 531.7 eV, which can be assigned to OD[23]. As the D₂ exposure increases from 0 to 240 L, the peak area of O_L decreases by 93% while that of OD increases. When D₂ exposure further increases to 300 L, XPS O 1s peak areas of O_L and OD remain almost unchanged, with their peaks located at 529.4 and 532.4 eV, respectively. Coincidently, the peak positions are consistent with those from the hydroxylated stripe-like MnO surface (Fig. 2a). The maximum degree of hydroxylation is determined to be 96%, indicating that MnOD_x should be formed. As shown in Supplementary Fig. 2, the ratio of total O to Mn signal is calculated to be 1.13, decreasing by ~13% compared with that of the as-prepared Mn₃O₄. This change of the O/Mn ratio suggests that the hydroxylation of grid-like Mn₃O₄ surface is accompanied by H₂O generation[26,27]. Notably, we find that H₂O begins to be generated as the hydroxylation degree of Mn₃O₄ is increased to ~50%.

The hydroxylation process of grid-like Mn₃O₄ surface was also investigated by HP-STM in H₂ at room temperature. The apparent height of the rim of grid-like Mn₃O₄ island is ~0.6 Å higher than the center area when the sample is exposed to H₂ (Fig. 2i). Inset in Fig. 2k

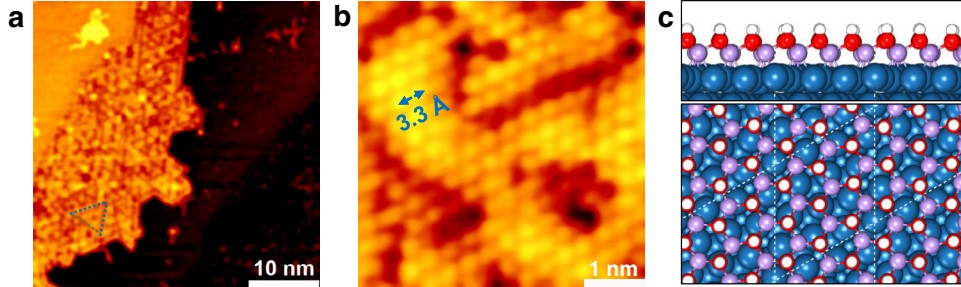

**Fig. 3 | Structure of the hydroxylated Mn oxide surface. a** STM image of a hydroxylated surface. **b** Atomic-resolution STM image of the hydroxylated surface. Scanning parameters: (**a**) $I_t = 0.100$ nA, $V_s = 0.999$ V; (**b**) $I_t = 0.210$ nA, $V_s = 0.004$ V.

**c** Proposed MnOH$_x$/Pt(111) surface structure. H: white; O: red; Mn: light violet; Pt: dark blue. The white dashed line indicates the boundary of the unit cell.

shows that the brighter rim is hexagonally symmetric (highlighted by a blue rhombus) with ~3.3 Å atomic distance. When H$_2$ partial pressure was increased to $2.5 \times 10^{-7}$ mbar, the brighter area continues to extend into the Mn$_3$O$_4$ island center with the hydroxylation front almost parallel to the edge of the island (Fig. 2i–n, for a time series see Supplementary Movie 2), which differs from the 1D spillover pathway on the stripe-like MnO island (Fig. 2c–h). In addition, the hydroxylation cannot take place on the 1.2 ML grid-like Mn$_3$O$_4$/Pt(111) surface (Supplementary Fig. 1c, d), indicating that the hydroxylation is also caused by hydrogen spillover from bare Pt(111) surface region to Mn$_3$O$_4$ islands.

STM images and LEED patterns of hydroxylated MnO and Mn$_3$O$_4$ surfaces are shown in Fig. 3 and Supplementary Fig. 3. Highlighted by blue dashed lines (Fig. 3a), the island surface presents hexagonal symmetry with regular triangles. The atomic distance is ~3.3 Å, exhibiting 19% lattice mismatch between the MnOH$_x$ overlayer and Pt(111) substrate. Since hydroxylation degree of MnO and Mn$_3$O$_4$ can both reach ~90%, the bright features in Fig. 3b can be assigned to OH species and the atomic model of MnOH$_x$/Pt(111) surface is thereby proposed in Fig. 3c. It should be noted that the hydroxylation of manganese oxide monolayers is not reversible. Zhang et al.[28] have shown that the desorption product of MnOH$_x$ film is H$_2$O instead of H$_2$.

**Kinetics of hydrogen spillover on MnO(001) and Mn$_3$O$_4$(001) surfaces**

Previous studies have shown that doping[29], molecular carriers[11,30], spectator molecules[25], interface length[17] and others can regulate the surface hydrogen migration amount, distance, and rate. However, the correlation between intrinsic properties of the oxide surface and the rate of hydrogen spillover has not been clearly understood. Figure 4a, b displays the normalized OD and O$_L$ contents of stripe-like MnO and grid-like Mn$_3$O$_4$ as a function of D$_2$ exposure amount. The normalized O$_L$ contents of stripe-like MnO as a function of D$_2$ exposure amount can be fitted using a *two parallel sites* model[31]: $\theta = 200.75 \times 0.001 \times e^{-0.001 \times n\,L} + 11.1 \times 0.07 \times e^{-0.07 \times n\,L}$ (n: D$_2$ exposure amount; L = $10^{-6}$ mbar·s). This means that two kinds of hydrogen diffusion pathways exist on the stripe-like MnO/Pt(111) surface. According to the in situ STM experiment, the hydrogen diffusion along the stripe is the dominant pathway compared with that across the stripe. Hydrogen spillover rate as a function of hydroxylation degree can be derived from the slope of the profile of normalized lattice oxygen contents as a function of D$_2$ exposure amount shown in Fig. 4a. When the exposure amount is <35 L (hydroxylation degree <70%), the hydrogen spillover rate remains nearly unchanged. With the H$_2$ exposure amount is >35 L, the rate of hydrogen spillover decreases rapidly with the increasing degree of hydroxylation. The normalized O$_L$ contents of grid-like Mn$_3$O$_4$ as a function of D$_2$ exposure amount can be fitted using a *single site* model[31]: $\theta = 76.5 \times 0.01 \times e^{-0.01 \times n\,L}$ (n: D$_2$ exposure amount; L = $10^{-6}$ mbar·s), implying that only

one hydrogen diffusion pathway exists. As indicated by the slope of profiles in Fig. 4b, the rate of hydrogen spillover keeps on decreasing with the increasing degree of hydroxylation. In addition, on the D$_2$ exposure traces, crossing points are observed at 16 L for MnO and 52 L for Mn$_3$O$_4$, indicating that hydrogen atoms diffuse faster on stripe-like MnO than on grid-like Mn$_3$O$_4$. We also studied the isotopic effect of spillover by using H$_2$. As shown in Supplementary Fig. 4, the hydroxylation extent of the MnO and Mn$_3$O$_4$ films in H$_2$ is slightly higher compared with the case in D$_2$. This indicates the existence of a normal kinetic isotopic effect ($k_H/k_D > 1$)[32,33].

In order to investigate the relationship between hydrogen spillover rate and H$_2$ partial pressure, in situ STM experiments were conducted. Firstly, stripe-like MnO overlayer was exposed to H$_2$ at room temperature. By comparing the lengths of the bright lines at different exposure time, the growth rates can be calculated, which represents the hydrogen spillover rates. The initial hydrogen spillover rates were calculated under different H$_2$ partial pressures, including $8.0 \times 10^{-9}$, $2.3 \times 10^{-8}$, $4.7 \times 10^{-8}$, $1.6 \times 10^{-7}$, and $3.0 \times 10^{-7}$ mbar H$_2$, and the logarithms of initial spillover rates vs. logarithms of $p_{H_2}$ are summarized in the orange line in Fig. 4c. The logarithms of hydrogen spillover rates show a half-order dependence on logarithms of H$_2$ partial pressure (gradient 0.45) at room temperature, suggesting that the hydrogen diffusion is the rate-determining step.

The hydrogen spillover rates of grid-like Mn$_3$O$_4$ in different H$_2$ partial pressures were investigated afterwards. By comparing the widths of the bright rim at different dosing times in the initial stage, the hydrogen spillover rates can be estimated. The initial hydrogen spillover rates were calculated in $9.3 \times 10^{-8}$, $1.9 \times 10^{-7}$, $5.3 \times 10^{-7}$, $7.9 \times 10^{-7}$, and $1.0 \times 10^{-6}$ mbar H$_2$ and the logarithms of initial spillover rates vs. logarithms of $p_{H_2}$ are shown in the dark cyan line in Fig. 4c. The fitted curve shares nearly the same gradient (0.47) as that of stripe-like MnO. Notably, the hydrogen diffusion rates on stripe-like MnO are four times faster than those on grid-like Mn$_3$O$_4$. This is probably accelerated by the 1D surface-lattice-confinement effect. The comparison of the spillover directions and rates on the two surfaces is illustrated in Fig. 4d. In addition, on both stripe-like MnO surface and grid-like Mn$_3$O$_4$ surface, hydrogen diffuses faster with the increasing H$_2$ partial pressure, which highlights the demanding of in situ characterization of catalysts under atmospheres.

**Theoretical insights into hydrogen spillover**

To gain insights into the difference of hydrogen diffusion on MnO and Mn$_3$O$_4$ surfaces, DFT calculations were carried out to obtain the energetics and electronic characters as shown in Fig. 5. According to our STM image data and the lattice mismatches[20,22], two kinds of MnO$_x$ ribbons on Pt(111), as shown in Figs. 1c, f, 5a, b, i.e., MnO/Pt(111) and Mn$_3$O$_4$/Pt(111), are utilized to simulate the local domains of the experimentally observed MnO and Mn$_3$O$_4$ monolayers.

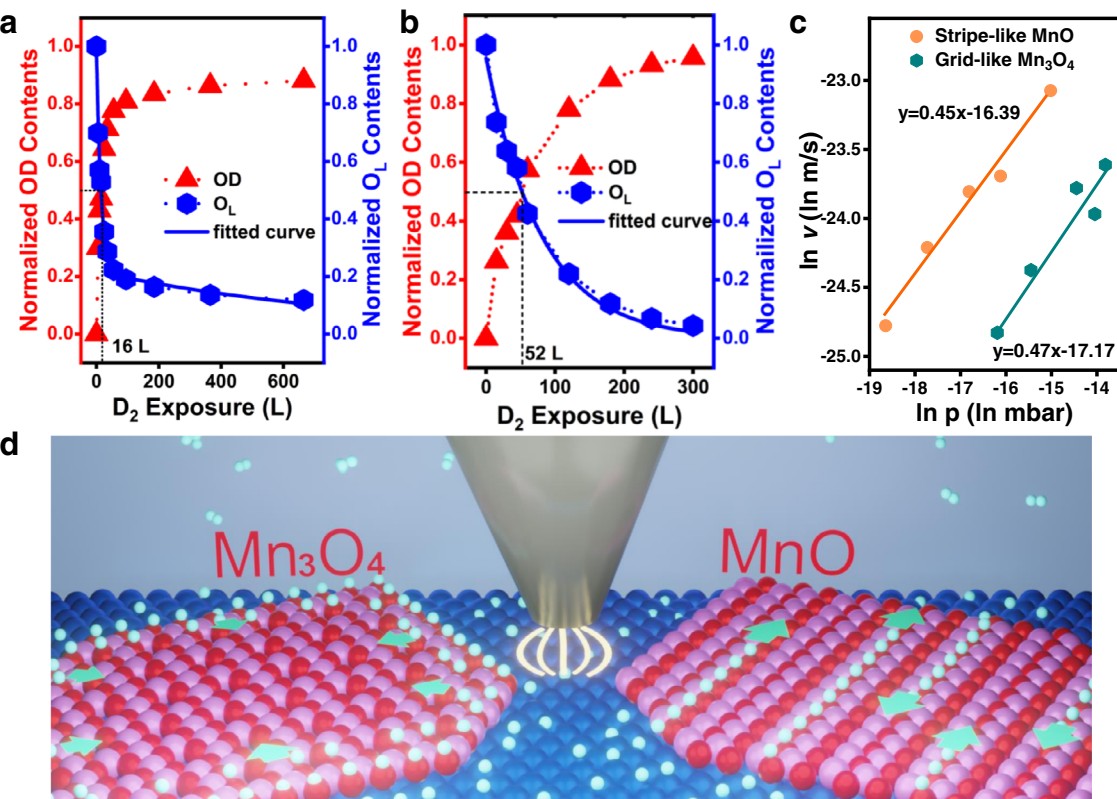

**Fig. 4 | Hydrogen spillover rates on MnO and Mn₃O₄.** OD and $O_L$ contents derived from the XPS O 1$s$ areas of (**a**) stripe-like MnO and (**b**) grid-like Mn₃O₄ surfaces with different amount of D₂ exposure. **c** Dependence of spillover rates of stripe-like MnO and grid-like Mn₃O₄ surfaces on H₂ partial pressure. The logarithms of spillover rates *vs.* logarithms of $p_{H_2}$. **d** Schematic of HP-STM using a STM tip to probe the hydrogen spillover on MnO and Mn₃O₄ surfaces in H₂ atmosphere. Pt: dark blue; Mn: light violet; O: red; H: cyan. Source data are provided as a Source Data file.

For the monolayer MnO/Pt(111) surface (Fig. 5a), due to the reconstruction induced by the MnO-Pt(111) mismatch there are two distinct pathways for hydrogen diffusion from one to another O sites (termed S$_n$, n is the designated number), including along the [01$\bar{1}$] direction (S1 → S2 → S3) and the [2$\bar{1}\bar{1}$] direction (S1 → S4 → S5). Figure 5c, Supplementary Fig. 5a and Supplementary Fig. 5b show that along the [01$\bar{1}$] direction the barriers for hydrogen diffusion from S1 to S2 and from S2 to S3 are nearly the same, i.e., 1.01 eV, while along the [2$\bar{1}\bar{1}$] direction hydrogen diffusion needs to overcome a barrier of 1.47 eV from S1 to S4 and 1.21 eV from S4 to S5. This indicates that hydrogen diffusion on MnO/Pt(111) is of a direction selectivity where the [01$\bar{1}$] direction is preferential with a barrier of -1.0 eV. We also consider a possibility of hydrogen diffusion via the Mn-H* intermediate but it is excluded due to its high barrier of 2.37 eV (Supplementary Fig. 6). Such results well explain the experimental observation that hydrogen unidirectionally diffuses on the MnO/Pt(111) surface (Fig. 2c–h), i.e., along the [01$\bar{1}$] direction and through a OH-to-OH mode. As for the Mn₃O₄/Pt(111) surface (Fig. 5b, c and Supplementary Fig. 5c), along the [01$\bar{1}$] direction the barrier for hydrogen diffusion across the Mn vacancy from S6 to S7 is 1.19 eV and that from S7 to S8 is 1.23 eV, both of which are higher than that of -1.0 eV for the preferential pathway on MnO/Pt(111). This means that hydrogen diffusion on MnO/Pt(111) is much easier than that on Mn₃O₄/Pt(111), agreeing well with our XPS and STM results (Fig. 4).

We further aim to understand the nature regarding the difference of hydrogen spilling over MnO/Pt(111) *vs.* Mn₃O₄/Pt(111). Figure 5d shows the *p*-orbital projected density of states (PDOS) of various O sites on MnO/Pt(111) and Mn₃O₄/Pt(111), where a upshift of O *p*-band center ($\varepsilon_p$) from −3.12 eV on MnO/Pt(111) to −2.61 eV on Mn₃O₄/Pt(111) is found. In some cases, the activity of O atom in oxides can be described by the *p*-band center, namely that the upshifting (more

positive) of $\varepsilon_p$ corresponds to a higher activity[34,35]. The movement of $\varepsilon_p$ just corresponds to a change of coordination number from O$_{4c}$ (4c denotes four-coordination) on MnO/Pt(111) to O$_{3c}$ on Mn₃O₄/Pt(111), implying an increased activity of O. Through the binding energy calculation, we find that the adsorption energy of H on MnO/Pt(111) is −0.39 eV while that on Mn₃O₄/Pt(111) is −1.13 eV, indicating a higher stability and then a harder diffusion for H* on Mn₃O₄/Pt(111) than on MnO/Pt(111). Therefore, we suggest that hydrogen diffusion depends on the O·H* stability which intrinsically lies in difference of the activity of surface O atoms as indicated by the *p*-band center or the coordination number. In addition, we deduce that a medium O-O distance along the 1D direction (lattice confinement) is favorable for H diffusion, such as 2.77 Å (Fig. 1c, f), which implies a geometric effect beyond the coordination number. The (001) facets of single crystal (abbreviation as sc) MnO and Mn₃O₄ were constructed to investigate the influence of the interaction between Pt substrate and MnO$_x$ monolayers toward the spillover rate. Comparing with the diffusion barriers of 1.23 eV on sc-MnO(001) and 1.57 eV on sc-Mn₃O₄(001) surfaces (Supplementary Fig. 7), we conclude that Pt substrate can promote H diffusion on the monolayer MnO$_x$ films via the strong MnO$_x$ and Pt interaction.

By extension, the effect of hydroxylation degree on H diffusion is further investigated. Supplementary Fig. 8 shows the H diffusion on the hydroxylated MnO$_x$/Pt surface (the model with one H atom adsorbed on the adjacent O site). For MnO, the barrier for H diffusion is 1.06 eV, which is slightly higher than that on H-free surface (1.01 eV). For Mn₃O₄, H diffusion needs to surmount a 1.24 eV barrier near to those on H-free surface (1.19 or 1.23 eV). This implies that when H coverage is low the effect of hydroxylation on H diffusion is trivial in the two film systems. However, if hydroxylation induces the geometric evolution from the tetragonal (t-MnO/Pt) to hexagonal (h-MnO/Pt)

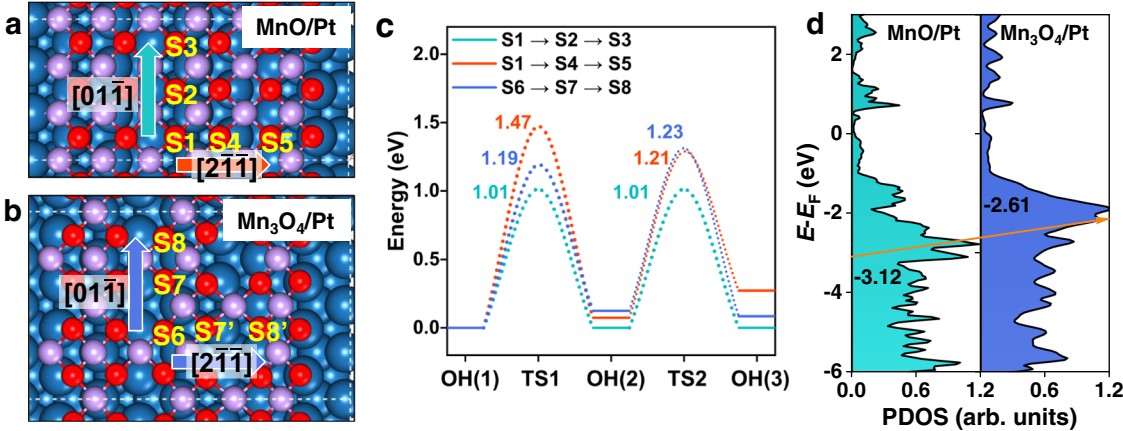

**Fig. 5 | Theoretical analysis on difference of hydrogen diffusion over the MnO and Mn₃O₄ monolayers.** Proposed monolayer (**a**) MnO and (**b**) Mn₃O₄ films supported on Pt(111) substrate. H: white; O: red; Mn: light violet; Pt: dark blue. Here, the characteristic surface O sites are denoted by yellow symbols and the hydrogen diffusion directions are indicated by colored arrows. **c** Potential energy diagram for O-H diffusion from site to site corresponding to (**a**) and (**b**). The zero energy level is relative to that of the first OH(1) state (i.e., H* on S1 and S6). The colored numbers denote the barriers for each elementary step. **d** Projected density of states (PDOS) of surface O sites (S1 and S6) in MnO/Pt(111) and Mn₃O₄/Pt(111), where the *p*-band centers of O are shown by the inserted numbers and their shifting direction is denoted by the orange arrow.

phases (Supplementary Fig. 8b), H diffusion would be harder with a barrier of 1.78 eV. We thus infer that the reduced hydrogen spillover rate with the increasing hydroxylation degree as observed in experiments (Fig. 4a, b) may stem from the structural evolution induced by hydroxylation.

It is generally accepted that molecules confined in 1D spaces, such as carbon nanotubes[36], zeolites[37], metal organic frameworks[38], and covalent organic frameworks[39], diffuse along the 1D channels with enhanced transport properties. By adjusting the size of the confinement space and the strength of the host-guest interaction, diffusion behavior of molecules can be effectively modulated. However, confined diffusion of molecules on an open space or surface has been rarely reported. One example is that molecules diffuse on TiO₂(110) surface preferentially along the shorter surface unit cell vector [001] direction[40-44]. As an open surface, stripe-like MnO(001) surface can regulate the hydrogen species to diffuse only along the shorter unit cell vector [01$\bar{1}$] direction, which derives from 1D surface-lattice-confinement effect. More importantly, hydrogen spillover rates are accelerated by the 1D surface-lattice-confinement effect, which has not been reported previously. During this process, the geometric effect plays a dominant role in H diffusion which further supports that the 1D surface-lattice-confinement effect accelerates H diffusion on monolayer MnO.

## Discussion

In summary, stripe-like MnO(001) and grid-like Mn₃O₄(001) monolayers are constructed on Pt(111) substrate. In situ HP-STM and XPS investigations show that the surface-lattice-confinement effect can regulate the hydrogen spillover directions and accelerate the spillover rates. DFT calculations indicate that this is intrinsically related to the different local surface geometries and coordination numbers of surface O sites in the two systems. These findings illustrate the effect of surface structure on the kinetics of hydrogen spillover in oxide systems. It deepens our understanding of the factors that influence the rate of hydrogen spillover, which in some cases is the rate-determine step during the hydrogenation or dehydrogenation process on oxide catalysts.

## Methods
### Sample preparations
Pt(111) substrate (MaTeck) used for manganese oxides growth was cleaned by cycles of 1.3 keV Ar⁺ sputtering and annealing at 800 K in O₂ atmosphere, followed by annealing at 1000 K in UHV. Manganese oxides were deposited by evaporating Mn shots in a Knudsen cell (Createc) in O₂ atmosphere. All gases were purified with liquid N₂ for more than 30 min before usage. The nominal MnOₓ coverage is obtained from the statistical STM images.

### Characterizations
All characterizations were conducted in two UHV systems. One consists of a sample preparation chamber and HP-STM (SPECS, Germany) with a base pressure $<3 \times 10^{-10}$ mbar. HP-STM uses a mechanically cut Pt-Ir tip. For in situ STM experiments, the imaging experiments were conducted at room temperature. STM images were obtained in the constant current mode and processed by SPIP (Image Metrology, Denmark). The other is an Omicron multiprobe system, which is equipped with a sample preparation chamber (base pressure $<5 \times 10^{-10}$ mbar), a spectroscopic chamber (base pressure $<3 \times 10^{-11}$ mbar), and a microscopic chamber (base pressure $<3 \times 10^{-10}$ mbar). The spectroscopic chamber is equipped with XPS (Omicron, NG DAR 400), and core-level spectra are acquired using Al Kα (hv = 1486.6 eV) radiation and a hemispherical energy analyzer (Omicron, EA 125 U7). XPS measurements were conducted after the as-prepared MnO/Pt(111) and Mn₃O₄/Pt(111) surfaces were exposed to different amount of D₂ at room temperature. The as-prepared MnO/Pt(111) surface was successively exposed to $1 \times 10^{-7}$ mbar D₂ for 50 s, $1 \times 10^{-7}$ mbar D₂ for 50 s, $1 \times 10^{-7}$ mbar D₂ for 50 s, $1 \times 10^{-7}$ mbar D₂ for 100 s, $1 \times 10^{-7}$ mbar D₂ for 100 s, $1 \times 10^{-7}$ mbar D₂ for 200 s, $5 \times 10^{-7}$ mbar D₂ for 80 s, $5 \times 10^{-7}$ mbar D₂ for 180 s, $5 \times 10^{-7}$ mbar D₂ for 360 s, and $1 \times 10^{-6}$ mbar D₂ for 300 s. The corresponding exposure amount of D₂ was 5, 10, 15, 25, 35, 55, 95, 185, 365, and 665 L. XPS measurements on Mn₃O₄/Pt(111) were conducted in two experiments. The as-prepared Mn₃O₄/Pt(111) surface was successively exposed to $5 \times 10^{-7}$ mbar D₂ for 30, 30, and 30 s. Another as-prepared Mn₃O₄/Pt(111) surface was successively exposed to $5 \times 10^{-7}$ mbar D₂ for 120, 120, 120, 120, and 120 s. Accordingly, the exposure amount of D₂ was 15, 30, 45, 60, 120, 180, 240, and 300 L. The normalized OD contents $= \frac{\text{Area}_{OD}}{\text{Area}_{OD} + \text{Area}_{O_L}}$; the normalized O$_L$ contents $= \frac{\text{Area}_{O_L}}{\text{Area}_{OD} + \text{Area}_{O_L}}$.

All XPS spectra were analyzed by CasaXPS software with a Linear background subtraction and Gaussian-Lorentzian fitting. The binding energy of O 1$s$ is corrected by using Pt 4$f$ as a reference.

## Computational parameters

Spin-polarized DFT calculations were implemented using a plane-wave basis set in the Vienna Ab-initio Simulation Packages (VASP 5.4)[45]. The exchange-correlation energy was treated using Perdew-Burke-Ernzerhof (PBE) functional within the generalized gradient approximation (GGA)[46]. The projected-augmented wave (PAW) pseudopotentials were utilized to describe the core electrons, and a cutoff energy of 400 eV was used for the plane-wave expansion[47]. The van der Waals (vdW) dispersion forces were corrected by the vdW-DF (optPBE) function, which showed highly accurate description for oxides[48]. An on-site Hubbard term $U_{eff} = U - J$ was added to address the open-shell $d$-electrons with 3.7 eV for Mn in the $MnO_x$/Pt(111) system[49]. The water-based reference state for the calculations to avoid incorrect description of the gas phase $O_2$ reference with standard DFT methods[48]. The energies and residual forces were converged to $10^{-5}$ eV and 0.02 eV·Å$^{-1}$, respectively. The searching of transition states (TSs) is through the climbing image nudged elastic band (CI-NEB) method[50].

## Model constructions

With consideration of the MnO/Pt(111) morphology and the lattice mismatch, a slab model of a $(5\sqrt{3} \times 3)$ rectangular supercell was used, where a monolayer $(6 \times 3)$ MnO(001)-like ribbon was supported on three Pt(111) metal layers (20 Pt atoms in each layer) (Fig. 1c). This MnO/Pt(111) model is near to a "3" truncation of the "2-3-3" sequence, i.e., the $(19 \times 1)$ reconstruction observed by STM[20]. The optimized lattice constants are 2.82 Å for p(1×1) Pt(111), which is about 1.02 times than the experimental value (2.77 Å) due to the systematic error from DFT calculation, and 2.97 Å for free-standing p(1×1) MnO(001) monolayer, respectively. Thus, the MnO ribbon on Pt(111) suffers from an about −5.1% compressive strain along the $[01\bar{1}]$ direction relative to the free-standing one. As for the monolayer $Mn_3O_4$/Pt(111) surface which is still ill-defined by STM, we first obtained a series of reconstructed films (bridged $Mn_{2c}$ transforms into quadruple $Mn_{4c}$, 2c denotes two-coordination) by optimizing the free-standing monolayer $Mn_3O_4$(001) (Supplementary Fig. 9a) with different sizes, including the (1×1), (2×2), and (3×3) supercells (Supplementary Fig. 9b–d). We find that the reconstruction modes for $Mn_{2c}$ are of difference among the three supercells, i.e., "A..B..C", "A..A..B..B", and "A..B..A" modes, respectively (Supplementary Fig. 9). We thus speculate that there may be more reconstruction modes and the experimental monolayer $Mn_3O_4$ should feature one of the $Mn_{2c}$ reconstructions. This motivates us to propose an $Mn_3O_4$ ribbon consisting of six columns of Mn and O atoms on a $(5\sqrt{3} \times 4)$ Pt(111) surface as shown in Fig. 1f, which is on the basis of the "A..B..A" reconstruction and whose $Mn_3O_4$ overlayer is of nearly 0.9% tensile strain along the $[01\bar{1}]$ direction. In addition, we utilize the (001) facets of single crystal MnO (Fm$\bar{3}$m, no. 225) and $Mn_3O_4$ (I4$_1$/amd, no. 141) to mimic the extreme situation of $MnO_x$ multilayers grown on Pt. For Brillouin zone integration, we employ (12 × 12 × 12) and (8 × 8 × 1) k-point grids within the Morkhorst-Pack scheme for Pt bulk and MnO(001) monolayer, respectively. For other films, facets, and hybrid $MnO_x$/Pt(111) interfaces, equivalent k-point grids are utilized. The vacuum layer is set as over 13 Å to avoid the spurious interation between slabs. In addition, the magnetism for the $MnO_x$/Pt(111) is simply set as the anti-ferromagnetic (AFM) ordering with consideration of its negligible influence on the tendency[51].

Notably, we find the interface matching is clearly different between Pt(111) and MnO vs. $Mn_3O_4$ monolayers, where the monolayer MnO exhibits stronger adhesion to the Pt substrate through more Mn-Pt bonding (Fig. 1c). We thus assume that there may be an intrinsic difference for H diffusion along the $[2\bar{1}\bar{1}]$ and $[01\bar{1}]$ directions for the MnO/Pt(111) surface due to the

reconstruction, whereas for $Mn_3O_4$/Pt(111) the two diffusion directions should be the same if excluding the edge effect limited by the ribbon model. We also note that such $Mn_3O_4$ monolayer is equivalent to the tetragonal MnO monolayer with a certain concentration of Mn defects, and there is an apparent difference on the coordination numbers of O, i.e., 3 for $Mn_3O_4$ and 4 for MnO, respectively.

## Formulae

(1) The $p$-band center $(\varepsilon_p)$[42] of active O site is defined as

$$\varepsilon_p = \frac{\int_{-\infty}^{\infty} n_p(\varepsilon)\varepsilon d\varepsilon}{\int_{-\infty}^{\infty} n_p(\varepsilon)d\varepsilon} \tag{1}$$

(2) The binding energy for H on the $MnO_x$/Pt(111) $(E_{ads})$ is calculated by

$$E_{ads}(H) = E_{H^*} - E_* - \frac{1}{2}E_{H_2} \tag{2}$$

where $E_{H^*}$ and $E_*$ are the total energies of the H-adsorbed and the pristine surfaces, respectively.

## Data availability

Relevant data supporting the key findings of this study are available within the paper and the supplementary information file. Source data are provided with this paper. Source data are provided as a Source Data file. Source data are provided with this paper.

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

## Acknowledgements

This work was financially supported by National Key R&D Program of China (2021YFA1502800 to Q.F.), National Natural Science Foundation of China (No. 91945302 to R.M., No. 21825203 to Q.F., No. 22288201 to Q.F., and No. 22272162 to R.M.), Photon Science Center for Carbon Neutrality, LiaoNing Revitalization Talents Program (XLYC1902117 to Q.F.), and the Dalian National Laboratory for Clean Energy (DNL) Cooperation Fund (DNL201907 to Q.F.). We thank the fruitful discussions with Dr. Hongbo Zhang.

## Author contributions

Q.F. and R.M. conceived the idea and directed the project. Y.L. carried out the STM experiments. R.Z. carried out the XPS experiments. L.L. carried out the theoretical simulations. Y.L., Y.W., and C.L. carried out the LEED experiments. All authors are involved to discuss the results and write the paper.

## Competing interests

The authors declare no competing interests.
