## [Peer Review File · Nature Communications]

Direct Observation of Accelerating Hydrogen Spillover via Surface-Lattice-Confinement EffectReviewer #1 (Remarks to the Author):

The manuscript is interesting and progresses the understanding of the role(s) of the support on the phenomenon of hydrogen spillover. That there is orientational spillover of hydrogen over oxide supports is a nice finding and deserves publication in Nature Communication.

I have one cautionary note: in the conclusion, far-reaching conclusions about the ability to accelerate spillover and enhance catalytic performance. Such promises are not made and cannot be made from the data presented in the manuscript. Such too far-reaching conclusions must be avoided.

A related remark that the authors may consider is that of the role of hydrogen pressure. Of course catalytic reactions are not done at the low pressures investigated in the manuscript; can the authors put their results into perspective of higher hydrogen pressures, which would enhance the relevance of the presented data.

Reviewer #2 (Remarks to the Author):

Hydrogen spillover is a well-known surface phenomenon encountered frequently in catalytic hydrogenation/hydrogenolysis reactions, but the mechanistic details of this processes still lack sufficient understanding. In this study, the authors successfully employed in situ HP-STM and XPS techniques to demonstrate the dynamics of hydrogen spillover on Pt(111)-supported manganese oxide monolayers in a quantitative manner and assessed the site requirements of hydrogen spillover via DFT calculations. The endeavors made by the authors are highly appreciated, and the findings of this study will definitely enrich the understanding of hydrogen spillover at atomic level. I would like to recommend this manuscript for publication in Nature Communications, if my following concerns about the content can be resolved.

1. In this study, the authors adopted the degree of hydroxylation to describe the hydrogen spillover on the manganese oxide monolayers. It is interesting to know whether this hydroxylation is reversible at the experimental condition, considering the fact that that hydrogen spillover is generally regarded as a reversible process.
2. For the spillover experiments shown in Figure 2, one can image that the hydrogen atoms diffuse on nearly bare lattice oxygen sites at the begin of the experiment, while they diffuse on surface hydroxyl sites when a high degree of hydroxylation is reached. Could the author compare the rates of hydrogen diffusion in these two cases?
3. Related with the above question, the DFT calculations shown in Figure 5 only considered the case of hydrogen diffusion on bare manganese oxide monolayers. How will the surface hydroxylation affect the diffusion rate?
4. Based on the DFT calculations, the authors concluded that both of the geometry and coordination number affect the rate of hydrogen spillover. I am afraid the authors did not examine the effect of the Pt(111) substrate on the electronic structure of the manganese oxide monolayers. It is generally accepted that the diffusion of hydrogen on reducible oxides via coherent proton-electron movements (Chem. Rev. 2012, 112, 2714–2738). Therefore, the rate of hydrogen spillover may also be influenced by the interaction between the Pt(111) substrate and the manganese oxide monolayer. It is suggested to calculate the barriers of hydrogen spillover for manganese oxide of multilayers and compare them with those for monolayers?
5. It is noticeable that the authors used H₂ for measuring the dependence of spillover rates on H₂ partial pressure (Figure 4c), while D₂ was used in other experiments. It is interesting to know whether an isotopic effect exists for the rates of hydrogen spillover.
6. At the end of the manuscript, the authors emphasized that hydrogen spillover rates

are accelerated by the 1D surface-lattice-confinement effect. I agree with the authors that the hydrogen spillover rates are affected by the geometry and coordination number of the surface sites, which can be attributable to the confinement of these sites on the surface. However, I am afraid that the authors did not provide compelling evidence to show the necessity or impact of one dimension.

7. In Line 205, a fitted equation was provided for describing the normalized lattice oxygen contents of stripe-like MnO as a function of D2 exposure amount. It seems that the term of D2 exposure amount was missed in this equation. It is also not clear how this equation is derived from a two parallel sites model, although the authors cited a reference for this model.

Reviewer #3 (Remarks to the Author):

This work reported an atomic-level characterizations of the hydrogen spillover on model MnOx/Pt surfaces through high-pressure STM, where a dynamic surface imaging of hydrogen transfer is captured. It is interesting to see that hydrogen species from Pt diffuses unidirectionally along the stripes on MnO(001) but in an isotropic pathway on Mn3O4(001) and hydrogen diffuses 4 times more rapidly on MnO than on Mn3O4 due to one-dimension surface-lattice-confinement effect. Further theoretical analysis shows the O-O distance as well as the coordination number of O plays a crucial role in the hydrogen diffusion. It is suitable for the publication on Nature Communications towards the readers focusing on the hydrogen spillover. Specific comments are listed below.

1) Is it possible to determine the rate of H spillover from Pt to MnOx?

2) In general, the activation energy as the DFT computed determines the reaction rate and whether it is surmounted depends on the reaction temperature. What is the relation between the proposed H spillover rate and the temperature? Like the onset temperature of H diffusion on different MnOx/Pt?

3) The O p-band center is not a common descriptor. Could some references be cited with an expanded discussion in the main text?

4) In discussion part, some perspectives regarding the hydrogen spillover rate and its catalytic application should be expanded to raise the paper's significance.

The words in italics are the Reviewers' comments.

The words in normal font are the author's responses.

These in red are the text of the revision in the revised manuscript.

Response to Reviewer 1:

Reviewer #1 (Remarks to the Author):

The manuscript is interesting and progresses the understanding of the role(s) of the support on the phenomenon of hydrogen spillover. That there is orientational spillover of hydrogen over oxide supports is a nice finding and deserves publication in Nature Communication.

Author reply: We thank the reviewer for her/his positive comments and recommendation of publication. Our point-by-point responses are listed below.

1. I have one cautionary note: in the conclusion, far-reaching conclusions about the ability to accelerate spillover and enhance catalytic performance. Such promises are not made and cannot be made from the data presented in the manuscript. Such too far-reaching conclusions must be avoided.

Author reply: We thank the reviewer for pointing out the inaccurate expression of the conclusion part. We have revised the **Discussion** part in **Line 380 of Page 15 in the revised main text:** "These findings illustrate the effect of surface structure on the kinetics of hydrogen spillover in oxide systems. It deepens our understanding of the factors that influence the rate of hydrogen spillover, which in some cases is the rate-determine step during the hydrogenation or dehydrogenation process on oxide catalysts."

2. A related remark that the authors may consider is that of the role of hydrogen pressure. Of course catalytic reactions are not done at the low pressures investigated in the manuscript; can the authors put their results into perspective of higher hydrogen pressures, which would enhance the relevance of the presented data.

Author reply: We agree with the reviewer about the important role of hydrogen pressure, and the kinetic analysis in Fig. 4c also proves this point. The similar difference may be expected under ambient reaction conditions based on reasonable extrapolation to higher pressure. However, when the MnO/Pt(111) and Mn₃O₄/Pt(111) samples were exposed to higher hydrogen pressures, the hydrogen would diffuse too fast for our HP-STM to capture the spillover processes on the two oxide surfaces for further distinguishing the difference between them.

Response to Reviewer 2:

Reviewer #2 (Remarks to the Author):

Hydrogen spillover is a well-known surface phenomenon encountered frequently in

catalytic hydrogenation/hydrogenolysis reactions, but the mechanistic details of this processes still lack sufficient understanding. In this study, the authors successfully employed in situ HP-STM and XPS techniques to demonstrate the dynamics of hydrogen spillover on Pt(111)-supported manganese oxide monolayers in a quantitative manner and assessed the site requirements of hydrogen spillover via DFT calculations. The endeavors made by the authors are highly appreciated, and the findings of this study will definitely enrich the understanding of hydrogen spillover at atomic level. I would like to recommend this manuscript for publication in Nature Communications, if my following concerns about the content can be resolved.

Author reply: We thank the reviewer for the positive evaluation of our work and her/his constructive suggestions. Our point-by-point responses are listed below.

1. In this study, the authors adopted the degree of hydroxylation to describe the hydrogen spillover on the manganese oxide monolayers. It is interesting to know whether this hydroxylation is reversible at the experimental condition, considering the fact that that hydrogen spillover is generally regarded as a reversible process.

Author reply: We thank the reviewer for this question. The hydroxylation of manganese oxide monolayers is not reversible. Another work of our group [ACS Catal. **12**, 11918-11926 (2022)] shows that the desorption product of MnOH_x film is H₂O instead of H₂. This has been discussed in **Line 213 of Page 9 in the revised main text:** “It should be noted that the hydroxylation of manganese oxide monolayers is not reversible. Zhang et al.²⁸ have shown that the desorption product of MnOH_x film is H₂O instead of H₂.”

2. For the spillover experiments shown in Figure 2, one can image that the hydrogen atoms diffuse on nearly bare lattice oxygen sites at the begin of the experiment, while they diffuse on surface hydroxyl sites when a high degree of hydroxylation is reached. Could the author compare the rates of hydrogen diffusion in these two cases?

Author reply: Indeed, the rate we measured is an apparent spillover rate, where at least three processes related to the H migration should be involved: (i) H₂ molecules dissociate on Pt, (ii) H atoms diffuse from Pt to MnO_x, (iii) H atoms diffuse on MnO_x surface. Hydrogen spillover rate at various hydroxylation degree can be derived from the slope of the normalized lattice oxygen contents as a function of D₂ exposure amount profiles in Fig. 4a, b.

For MnO, the changes of the rate can be divided into two stages. When the exposure amount is less than 35 L (hydroxylation degree less than 70%), the hydrogen spillover rate remains nearly unchanged. With the H₂ exposure amount more than 35 L, the rate of hydrogen spillover decreases rapidly with the increasing hydroxylation degree. For Mn₃O₄, the rate of hydrogen spillover keeps on decreasing with the increasing hydroxylation degree. Overall, hydrogen diffuses more slowly on more strongly hydroxylated MnO and Mn₃O₄ films while the dependence of hydrogen spillover on the degree of hydroxylation shows certain difference in both cases.

The relevant discussion was added in **Line 229 of Page 9 in the revised main text**: “Hydrogen spillover rate as a function of hydroxylation degree can be derived from the slope of the profile of normalized lattice oxygen contents as a function of D₂ exposure amount shown in Fig. 4a. When the exposure amount is less than 35 L (hydroxylation degree less than 70%), the hydrogen spillover rate remains nearly unchanged. With the H₂ exposure amount more than 35 L, the rate of hydrogen spillover decreases rapidly with the increasing degree of hydroxylation.”

And **Line 238 of Page 10**: “As indicated by the slope of profiles in Fig. 4b, the rate of hydrogen spillover keeps on decreasing with the increasing degree of hydroxylation.”

3. Related with the above question, the DFT calculations shown in Figure 5 only considered the case of hydrogen diffusion on bare manganese oxide monolayers. How will the surface hydroxylation affect the diffusion rate?

Author reply: We thank the reviewer for pointing this out. We put one H atom on the surface O sites to mimic the hydroxylated surfaces and compare the H diffusion adjacent to the OH. The results are shown below and updated in the revised Supplementary Fig. 8. For MnO, the barrier for H diffusion is 1.06 eV, which is slightly higher than that on H-free surface (1.01 eV). For Mn₃O₄, H diffusion needs to surmount a 1.24 eV barrier near to those on H-free surface (1.19 or 1.23 eV). This implies that when H coverage is low the hydroxylation effect on H diffusion is trivial in the two film systems. However, if the hydroxylation induces the geometric evolution from the tetragonal to hexagonal phases, H diffusion would be harder with a barrier of 1.78 eV. The higher barrier of H diffusion may correspond to the reduced hydrogen spillover rate observed in experiments (Fig. 4a, b) when the degree of hydroxylation is higher e.g., 70% or the occurrence of the surface reconstruction to hexagonal phases.

The relevant discussion was added in **Line 345 of Page 14 in the revised main text and Supplementary Fig. 8 in the revised SI**: “By extension, the effect of hydroxylation degree on H diffusion is further investigated. Supplementary Fig. 8 shows the H diffusion on the hydroxylated MnO_x/Pt surface (the model with one H atom adsorbed on the adjacent O site). For MnO, the barrier for H diffusion is 1.06 eV, which is slightly higher than that on H-free surface (1.01 eV). For Mn₃O₄, H diffusion needs to surmount a 1.24 eV barrier near to those on H-free surface (1.19 or 1.23 eV). This implies that when H coverage is low the effect of hydroxylation on H diffusion is trivial in the two film systems. However, if hydroxylation induces the geometric evolution from the tetragonal (t-MnO/Pt) to hexagonal (h-MnO/Pt) phases (Supplementary Fig. 8b), H diffusion would be harder with a barrier of 1.78 eV. We thus infer that the reduced hydrogen spillover rate with the increasing hydroxylation degree as observed in experiments (Fig. 4a, b) may stem from the structural evolution induced by hydroxylation.”

Supplementary Fig. 8. H diffusion on the hydroxylated surfaces of tetragonal MnO/Pt (t-MnO/Pt) and Mn₃O₄/Pt, and the bare surface of hexagonal MnO/Pt (h-MnO/Pt). (a) Potential energy diagram of H diffusion from one to another O sites. The barriers are calculated based on the H diffusion pathways along the arrow direction. **(b)** Configurations of the corresponding transition states. S1 and S2 represent two different O sites.

4. Based on the DFT calculations, the authors concluded that both of the geometry and coordination number affect the rate of hydrogen spillover. I am afraid the authors did not examine the effect of the Pt(111) substrate on the electronic structure of the manganese oxide monolayers. It is generally accepted that the diffusion of hydrogen on reducible oxides via coherent proton-electron movements (Chem. Rev. 2012, 112, 2714–2738). Therefore, the rate of hydrogen spillover may also be influenced by the interaction between the Pt(111) substrate and the manganese oxide monolayer. It is suggested to calculate the barriers of hydrogen spillover for manganese oxide of multilayers and compare them with those for monolayers?

Author reply: We thank the reviewer for this interesting question. We agree with that the Pt substrate plays a role in surface chemistry of the MnO_x ultrathin films, including both the electronic and geometric effects due to the interface adhesion, which then influences the H diffusion. Here, we construct the (001) facets of single crystal MnO and Mn₃O₄ to simulate the extreme situation of multilayers, i.e., neglecting the Pt substrate effect. As shown in Supplementary Fig. 7, the barriers of H diffusion on MnO(001) and Mn₃O₄(001) are 1.23 and 1.57 eV, respectively, which surpass those on monolayer MnO/Pt (1.01 eV) and Mn₃O₄/Pt (~1.2 eV). Therefore, we can say that Pt substrate can promote H diffusion on monolayer MnO_x films compared to the multilayers via the strong MnO_x and Pt interaction.

The relevant discussion was added in **Line 338 of page 14 in the revised main text and Supplementary Fig. 7 in the revised SI**: “The (001) facets of single crystal (abbreviation as sc) MnO and Mn₃O₄ were constructed to investigate the influence of the interaction between Pt substrate and MnO_x monolayers towards the spillover rate. Comparing with the diffusion barriers of 1.23 eV on sc-MnO(001) and 1.57 eV on sc-Mn₃O₄(001) surfaces (Supplementary Fig. 7), we conclude that Pt substrate can promote

H diffusion on the monolayer MnO_x films via the strong MnO_x and Pt interaction.”.

And **Line 370 of Page 15 in the revised main text**: “During this process, the geometric effect plays a dominant role in H diffusion which further supports that the 1D surface-lattice-confinement effect accelerates H diffusion on monolayer MnO .”

Supplementary Fig. 7. H diffusion on the (001) facets of single crystal (abbreviation as sc) MnO and Mn_3O_4 . (a) Potential energy diagram of H diffusion from one to another O sites. The barriers are calculated based on the H diffusion pathways along the arrow direction. (b) Configurations of the corresponding transition states. S1 and S2 represent two different O sites.

5. It is noticeable that the authors used H_2 for measuring the dependence of spillover rates on H_2 partial pressure (Figure 4c), while D_2 was used in other experiments. It is interesting to know whether an isotopic effect exists for the rates of hydrogen spillover.

Author reply: We thank the reviewer for this nice suggestion. We have performed additional experiments to investigate the hydroxylation process of MnO and Mn_3O_4 films in H_2 via XPS. As shown in the revised Supplementary Fig. 4, the spillover rate on MnO and Mn_3O_4 films in H_2 is slightly faster than those in D_2 , which indicates that the kinetic isotopic effect does exist. This isotopic effect is probably due to the molecular weight difference between hydrogen and deuterium atoms. This result and its discussion have been added in **Line 243 of Page 10 in the revised main text and Supplementary Fig. 4 in the revised SI**: “We also studied the isotopic effect of spillover by using H_2 . As shown in Supplementary Fig. 4, the hydroxylation extent of the MnO and Mn_3O_4 films in H_2 is slightly higher compared with the case in D_2 . This indicates the existence of a normal kinetic isotopic effect ($k_{\text{H}}/k_{\text{D}} > 1$)^{32, 33}”

Supplementary Fig. 4 Hydroxylation process of MnO and Mn₃O₄ films in D₂ and H₂ at room temperature. OD and OH contents derived from the XPS O 1s areas of (a) stripe-like MnO and (b) grid-like Mn₃O₄ surfaces with different D₂ and H₂ exposure.

6. At the end of the manuscript, the authors emphasized that hydrogen spillover rates are accelerated by the 1D surface-lattice-confinement effect. I agree with the authors that the hydrogen spillover rates are affected by the geometry and coordination number of the surface sites, which can be attributable to the confinement of these sites on the surface. However, I am afraid that the authors did not provide compelling evidence to show the necessity or impact of one dimension.

Author reply: We apologize for the confusion. On stripe-like MnO, the barriers of hydrogen diffusion along the $[01\bar{1}]$ direction (1.01 eV) is much lower than that along the $[2\bar{1}\bar{1}]$ direction (1.47 eV) (Fig. R1c). This indicates a direction selectivity of hydrogen diffusion on MnO/Pt(111) surface. The hydrogen spillover thus exhibits 1D path (Fig. R1a, b). On grid-like Mn₃O₄, the barriers of hydrogen diffusion along the $[01\bar{1}]$ direction and $[2\bar{1}\bar{1}]$ direction is the same (1.23 eV). (Fig. R1f). The hydrogen spillover thus exhibits 2D path (Fig. R1d, e). Meanwhile, the hydrogen atom migrates faster on MnO than on Mn₃O₄ (Fig. R1g). Therefore, the 1D stripes on MnO regulate the hydrogen atoms to diffuse only along the $[01\bar{1}]$ direction and accelerate the diffusion rate.

The main text is revised in **Line 366 of Page 15**: “As an open surface, stripe-like MnO(001) surface can regulate the hydrogen species to diffuse only along the shorter unit cell vector $[01\bar{1}]$ direction, which derives from 1D surface-lattice-confinement effect.”

Fig. R1 STM image of hydrogen spillover on (a, b) stripe-like MnO and (d, e) grid-like Mn₃O₄. Proposed monolayer (c) MnO and (f) Mn₃O₄ films supported on Pt(111) substrate. (g) Dependence of spillover rates of stripe-like MnO and grid-like Mn₃O₄ surfaces on H₂ partial pressure. The logarithms of spillover rates vs. logarithms of p_{H_2} .

7. In Line 205, a fitted equation was provided for describing the normalized lattice oxygen contents of stripe-like MnO as a function of D₂ exposure amount. It seems that the term of D₂ exposure amount was missed in this equation. It is also not clear how this equation is derived from a two parallel sites model, although the authors cited a reference for this model.

Author reply: We thank the reviewer for pointing this out. The fitted equation describing the normalized lattice oxygen contents of MnO as a function of D₂ exposure amount has been revised in **Line 224 of Page 9 in the main text**: “ $\theta = 200.75 \times 0.001 \times e^{-0.001 \times nL} + 11.1 \times 0.07 \times e^{-0.07 \times nL}$ (n: D₂ exposure amount; L = 10⁻⁶ mbar·s).”

The fitted equation of Mn₃O₄ has also been revised in **Line 237 of Page 9 in the main text**: “ $\theta = 76.5 \times 0.01 \times e^{-0.01 \times nL}$ (n: D₂ exposure amount; L = 10⁻⁶ mbar·s)”

Steady-state isotopic-transient kinetic analysis (SSITKA) was used as a source of reference to gain insight into the hydroxylation mechanism. The changes of the normalized OD and O_L contents can be regarded as an analogy to the SSITKA experiments, in which the signal of the OD group from the MnO_x surface was regarded as one isotope while the signal of O_L contents was taken as the other isotope. The step-decay of the O_L content was used to follow the hydroxylation process on the D₂ exposure traces.

The model of transient responses for an irreversible-reaction mechanism consisting of a single pool, two parallel pools, two series pools, etc. Fig. R2 (adopted from Ref. 31) summarizes the profiles of the transient response for three catalyst-surface mechanisms.

The *single pool* model exhibits single-exponential behavior: $F^P(t) = \exp(-t/\bar{\tau}^A)$, and two

parallel pools model exhibits multiexponential behavior: $F^P(t) = \bar{x}_1^A \exp(-t/\tau_1^A) + \bar{x}_2^A \exp(-$

$$t/\tau_2^A), \bar{x}_1^A = \frac{\bar{N}_1^A}{\bar{N}_1^A + \bar{N}_2^A}, \bar{x}_2^A = 1 - \bar{x}_1^A.$$

In Fig. 4a, the normalized lattice oxygen contents of stripe-like MnO as a function of D₂ exposure amount exhibits multiexponential behavior and can be fitted using the bi-exponential equation perfectly.

In Fig. 4b, the normalized lattice oxygen contents of grid-like Mn₃O₄ as a function of D₂ exposure amount exhibits single-exponential behavior and can be fitted using the single-exponential equation perfectly.

Fig. R2. Typical transient response for three catalyst-surface mechanisms. [Ref. 31]

Response to Reviewer 3:

Reviewer #3 (Remarks to the Author): This work reported an atomic-level characterizations of the hydrogen spillover on model MnO_x/Pt surfaces through high-pressure STM, where a dynamic surface imaging of hydrogen transfer is captured. It is interesting to see that hydrogen species from Pt diffuses unidirectionally along the stripes on MnO(001) but in an isotropic pathway on Mn₃O₄(001) and hydrogen diffuses 4 times more rapidly on MnO than on Mn₃O₄ due to one-dimension surface-lattice-confinement effect. Further theoretical analysis shows the O-O distance as well as the coordination number of O plays a crucial role in the hydrogen diffusion. It is suitable for the publication on Nature Communications towards the readers focusing on the hydrogen spillover. Specific comments are listed below.

Author reply: We thank the reviewer for the positive comments on our work. The point-to-point responses are listed below.

1. *Is it possible to determine the rate of H spillover from Pt to MnO_x?*

Author reply: We thank the reviewer for this question. Indeed, the spillover rate that we observed is an apparent rate where at least three processes related to the H migration should be involved: i) H₂ molecules dissociate on Pt; ii) H atoms diffuse from Pt to MnO_x;

iii) H diffuse on MnO_x surface. Thus, we have not distinguished the three aspects but focus on the overall result about the H spillover on the two MnO_x surfaces.

2. In general, the activation energy as the DFT computed determines the reaction rate and whether it is surmounted depends on the reaction temperature. What is the relation between the proposed H spillover rate and the temperature? Like the onset temperature of H diffusion on different MnO_x/Pt?

Author reply: We thank the reviewer for raising this question. We did all the experiments at room temperature (RT) and thus infer that the barrier for H diffusion can be overcome under this condition. The relationship between H spillover rate and temperature can be determined by carrying out experiments under different temperatures, which is, however, beyond our equipment conditions. Till now, we cannot determine the onset temperature for H diffusion on different MnO_x/Pt surfaces, which demands high pressure STM at variable temperatures.

3. The O p-band center is not a common descriptor. Could some references be cited with an expanded discussion in the main text?

Author reply: We thank the reviewer for this nice comment. We have cited the relevant references in **Line 324 of Page 13** in the revised main text and illustrates this descriptor: “In some cases, the activity of O atom in oxides can be described by the p-band center, namely that the upshifting (more positive) of ϵ_p corresponds to a higher activity^{34,35}”

4. In discussion part, some perspectives regarding the hydrogen spillover rate and its catalytic application should be expanded to raise the paper's significance.

Author reply: We thank the reviewer for the kind suggestion. Hydrogen spillover has been demonstrated and utilized in a variety of H-involving processes, included but not limited to hydrogen storage, hydrogenations, etc. In many cases, the diffusion of spilt hydrogen atoms is determined to be the rate-determining step in hydrogenation reactions. In this case, the hydrogen spillover rate will accelerate the reactivity rate. Its catalytic applications are summarized in the **Introduction** part. Further, we have revised the **Discussion** part in **Line 380 of Page 15 in the revised main text** to clarify the significance of our work: “These findings illustrate the effect of surface structure on the kinetics of hydrogen spillover in oxide systems. It deepens our understanding of the factors that influence the rate of hydrogen spillover, which in some cases is the rate-determine step during the hydrogenation or dehydrogenation process on oxide catalysts.”

Reviewer #2 (Remarks to the Author):

My concerns have been fully addressed by the authors, and I would like to recommend its publication in Nature Communications.

I fully agree with the suggestions made by Reviewer #1, and I think the authors have made an appropriate revision based on these suggestions.

Reviewer #3 (Remarks to the Author):

Authors have well responded my comments. I have no additional comments.

The words in italics are the Reviewers' comments.

The words in normal font are the author's responses.

These in red are the text of the revision in the revised manuscript.

Response to Reviewer 1:

Reviewer #2 (Remarks to the Author):

My concerns have been fully addressed by the authors, and I would like to recommend its publication in Nature Communications.

I fully agree with the suggestions made by Reviewer #1, and I think the authors have made an appropriate revision based on these suggestions.

Author reply: We thank the reviewer for approving our revisions and recommending the acceptance of our manuscript for publication.

Reviewer #3 (Remarks to the Author):

Authors have well responded my comments. I have no additional comments.

Author reply: We thank the reviewer very much for the positive comments on our revised manuscript.